# Prevalence of hepatitis B and C virus infections in Lao People's Democratic Republic: The first national population-based cross-sectional survey

Shinsuke Miyano[1]*, Chansay Pathammavong[2], Yasunori Ichimura[1], Masaya Sugiyama[3], Kongxay Phounphenghack[2], Chankham Tengbriacheu[4], Bouaphane Khamphaphongphane[5], Phonethipsavanh Nouanthong[6], Lauren Franzel[7], Tae Un Yang[7], Hendrikus Raaijimakers[8], Tomomi Ota[1], Masafumi Funato[1], Kenichi Komada[1], Masahiko Hachiya[1]

1 Bureau of International Health Cooperation and WHO Collaborating Center for Health Systems Development, National Center for Global Health and Medicine, Shinjuku, Tokyo, Japan, 2 National Immunization Program, Mother and Child Health Center, Ministry of Health, Lao People's Democratic Republic (Lao PDR), Vientiane Capital, Lao PDR, 3 Genome Medical Science Project, The Research Center for Hepatitis and Immunology, National Center for Global Health and Medicine, Ichikwa, Chiba, Japan, 4 Mother and Child Health Center, Ministry of Health, Vientiane Capital, Lao PDR, 5 National Center Laboratory and Epidemiology, Ministry of Health, Vientiane Capital, Lao PDR, 6 Institute Pasteur du Laos, National Immunization Technical Advisory Group, Ministry of Health, Vientiane Capital, Lao PDR, 7 Vaccine-Preventable Diseases and Immunization Team, WHO Lao PDR, Vientiane Capital, Lao PDR, 8 Health and Nutrition Section, UNICEF, Vientiane Capital, Lao PDR

* s-miyano@it.ncgm.go.jp

**Data Availability Statement:** All the relevant data are within the paper and its Supporting Information files.

## Abstract

Population-based seroprevalence of chronic hepatitis B and C infections has not been examined in Lao People's Democratic Republic (PDR). Therefore, this study aimed to estimate the seroprevalence of these infections in the general population of Lao PDR and perform subgroup analysis. A nationwide seroprevalence survey was conducted in Lao PDR in June 2019 using the multistage cluster sampling method. Dried blood spot samples were collected onto Whatman™ 903 filter paper by finger prick. A chemiluminescent microparticle immunoassay was used to measure the levels of hepatitis B surface antigen (HBsAg) and hepatitis C antibody (HCV-Ab). Samples in which the HBsAg level was above 0.05 IU/ml and HCV-Ab was above the signal/cutoff ratio of 1.0 were considered positive based on comparisons with the relative light unit value of a calibration sample. A total of 1,927 samples (male: 47.3%, mean age: 23.0 years) were included in the analysis. The prevalence was estimated to be 4.2% (95% confidence interval [CI]: 2.7–6.3) for HBsAg and 1.6% (95% CI: 0.5–5.3) for HCV-Ab. Multivariable analysis revealed that those aged 20–24 years (adjusted odds ratio (AOR): 2.3, 95% CI: 1.1–4.6), those aged 25–29 years (AOR: 2.7, 95% CI: 1.3–5.6), those from the Northern region (AOR: 2.8, 95% CI: 1.2–6.6), and those who were Khmu (AOR: 3.6, 95% CI: 2.0–6.8) or Hmong (AOR: 5.0, 95% CI: 3.3–7.5) were significantly more likely to be positive for HBsAg. Although there were no statistically significant differences in the HCV-Ab prevalence according to each variable, males (2.9%, 95% CI: 0.7–10.7), those aged ≥40 years (6.1%, 95% CI: 2.1–16.8), and those from the Southern

**Funding:** This work was supported by the National Center for Global Health and Medicine (NCGM) Intramural Research Fund, Japan [grant numbers: 25-8, 19A01 and 22A01], and the Grant for the National Immunization Program, Ministry of Health, Lao PDR [FY2019]. However, the funders had no role in study design, data collection, analysis, decision to publish, or preparation of the manuscript.

**Competing interests:** The authors declare that they have no conflict of interest.

region (3.3%, 95% CI: 0.6–15.3) tended to have a higher prevalence. This novel population-based survey found differences in the prevalence of chronic hepatitis B and hepatitis C virus infections in Lao PDR according to sex, age group, region, and ethnicity; however, the results of this study should be confirmed in future studies, and relevant responses tailored for each target also need to be determined to control the transmission of hepatitis B and C infections.

## Introduction

Over 354 million people are affected by chronic viral hepatitis B (HBV) and C (HCV) infections, which are the root causes of liver cancer [1–4]. In 2016, the World Health Organization (WHO) estimated that 19 million people would die of hepatitis by 2030 if no additional efforts were made against HBV and HCV [5]. Thus, the Global Health Sector Strategy on viral hepatitis 2016–2021 appealed for the elimination of viral hepatitis by 2030 [6]. Based on modeling analysis, combining prevention and treatment is a key to global hepatitis elimination [5,7]. Prevention can reduce new infections, and treatment can prevent transmission and poor hepatitis-related outcomes in the short- and medium-terms [8,9].

Treatments for chronic HBV and HCV are currently available globally. Although there is no licensed vaccine against HCV, the HBV vaccine has become affordable and available in many countries. However, further efforts are needed to increase the coverage of the timely birth dose of the HBV vaccine [7,10,11]. Access to testing and treatment for chronic HBV and HCV infections also remains limited, mainly in developing countries [7,10,11]. Since those countries are often less likely to have epidemiological data, it is difficult to estimate the demand for and assess the coverage of service delivery. Seroprevalence surveys could provide baseline information from those countries for the expansion of services [12–17].

Lao People's Democratic Republic (PDR) has been regarded as a hyperendemic country for HBV infection and is considered a priority country by the WHO, despite a lack of population-based prevalence data. The 8.7% prevalence of HBV surface antigen (HBsAg) obtained from blood donors in three provinces in 2007 has been used broadly as the official estimation of chronic HBV infection in Lao PDR [18,19].

The National Immunization Program introduced the HBV vaccine into the routine immunization program in 2001 (at 6, 10, and 14 weeks after birth) and initiated birth dosing at referral hospitals in the capital city (2004), and rural hospitals (2006). The third-dose coverage rate was around 50% in 2007, but gradually increased to 92% in 2019, while the birth dose coverage remained low (66% in 2019) [20]. The first nationwide survey for HBsAg prevalence in children and their mothers was conducted in 2012, showing that estimated prevalence was 1.7% (95% CI: 0.8%–2.6%) in children and 2.9% (95% CI: 1.7%–4.2%) in their mothers, which was much lower than the broadly used estimation [18,21]. However, no population-based studies have investigated the prevalence of HBV in the general population [19,21–33].

To the best of our knowledge, no national representative seroprevalence data on HCV infection in Lao PDR exist, although some studies have indicated the prevalence of HCV infection among blood donors (1.1%) [19], healthcare workers (3.9%) [31], and female workers in garment factories (1.8%) [26].

Thus, we used samples from the nationwide seroprevalence survey for measles and rubella in 2019 (submission in progress) to estimate the prevalence of HBV and HCV infections in the general population in Lao PDR.

## Materials and methods

We conducted a nationwide seroprevalence survey in 2019 to estimate the immunity level against measles and rubella using dried blood spot (DBS) samples collected from randomly selected participants. We examined hepatitis B surface antigen (HBsAg) and anti-HCV antibody (HCV-Ab) using the remaining DBS samples.

### Sample size calculation

The sample size ($n$) was calculated as $n = Z^2 \times p(1-p)DEFF/(d^2 \times RR)$ where $Z$ = significance level for 95% confidence = 1.96; $p$ = expected prevalence = 0.1; $DEFF$ = design effect = 1.6; $d$ = precision = +/−0.05 to +/−0.06; $RR$ = response rate = 0.99. We assumed a 10% prevalence for HBsAg and HCV-Ab based on the results of past studies in Lao PDR [19,21–33] and calculated a required sample size of 1,872.

### Survey design and sampling strategies

We utilized a three-stage random cluster sampling design [34,35]. For the first stage, 26 of 148 districts were randomly selected by applying probability proportionate to size (PPS) sampling based on the 2005 population census. For the second stage, two villages were randomly selected from each district by PPS sampling. For the third stage, 42 participants (including 8 participants in the 1–2 years, 6 in the 5–14 years, 6 in the 15–19 years, 16 in the 20–39 years, and 6 in the ≥40 years age groups) were randomly selected from the households list using a paper-based lottery in each village. A survey team conducted a brief face-to-face interview with the participants to obtain demographical information and collected blood by finger prick [36].

### Laboratory examination for HBsAg and HCV-Ab

A small amount of blood was placed onto Whatman 903 filter paper protein saver (Whatman, Maidstone, Kent, UK) by finger prick and air-dried for at least 60 minutes [36]. The filter papers were sealed in plastic bags with desiccant at ambient temperature and transported to the National Center for Global Health and Medicine, Japan, within a week after collection. Blood samples were extracted from the DBS samples by punching six blood-stained circles (diameter, 3 mm) and eluting overnight in 500 μL of phosphate-buffered saline (pH 7.2). The eluates were tested for HBsAg and HCV-Ab using a chemiluminescent microparticle immunoassay (Architect i2000SR; Abbott Diagnostics, IL). An automated system was used to detect each sample's relative light unit (RLU) value. Positive samples were considered based on comparisons with the RLU value of a calibration sample [30].

### Data entry and statistical analysis

All the collected data were double-entered and cleaned on a Microsoft Excel 2017 spreadsheet. Statistical analysis was conducted using STATA version 14 (Stata Corp., College Station, TX, USA). All estimates and standard errors were calculated using a multistage clustered sampling design, considering the weight of each sample to elicit representative and unbiased results. Bivariable and multivariate analyses were performed to assess risk factors for HBV infection but not for HCV infection because of an insufficient number of samples were HCV-Ab positive [37,38]. Independent variables significantly associated with a positive HBsAg result in the bivariate analysis (p < 0.2) were included in the multivariate analysis. Variables with high multicollinearity (variance inflation factor: VIF > 10) were excluded from the final multivariate

logistic regression model. Backward stepwise selection was applied for variables with a significance level of 0.05. A P-value of <0.05 was considered statistically significant.

## Ethical considerations

Written informed consent was obtained from all the selected participants or their parents or legal representatives. Participants' names were not recorded. The research proposal was approved by the National Center for Global Health and Medicine, Japan (NCGM-3038) and the Ministry of Health, Lao PDR (06/NECHR).

## Results

The survey teams visited all 52 selected villages in the 26 districts between 3rd and 14th of June 2019 and completed blood sampling in 2,043 people. After excluding samples due to a lack of demographic information or improper blood sampling, 1,927 samples underwent laboratory examination (103.0% of the required sample size). The mean age was 23.0 years, ranging from 1 to 89 years. Males accounted for 47.3% of all selected subjects. Among the samples, 27.2% were from the Northern region, 49.6% from the Central region, and 23.1% were from the Southern region. By ethnicity, 58.2% were Laoloum, 15.2% were Khmu, 16.8% were Hmong, and 9.8% were from other groups. Approximately 20% of the participants had not received any education, while others completed primary school (38.9%) and secondary school and higher (40.7%) (Table 1).

**Table 1. Demographics of survey participants (N = 1,927).**

|  | n (%) |
|---|---|
| **Age (mean, years)** | 23.0 |
| **Sex** |  |
| Female | 1,016 (52.7) |
| Male | 1,001 (47.3) |
| **Age (years)** |  |
| 1–9 | 521 (27.0) |
| 10–19 | 448 (23.2) |
| 20–29 | 307 (15.9) |
| 30–39 | 282 (14.6) |
| 40– | 369 (19.1) |
| **Place of residence (region)** |  |
| Northern | 525 (27.2) |
| Central | 956 (49.6) |
| Southern | 446 (23.1) |
| **Ethnicity** |  |
| Laoloum | 1,121 (58.2) |
| Khmu | 293 (15.2) |
| Hmong | 322 (16.8) |
| Others | 191 (9.8) |
| **Education history**[*] |  |
| None | 353 (20.4) |
| Primary | 672 (38.9) |
| Secondary and higher | 703 (40.7) |

[*]Missing information of 199 individuals.

## Seroprevalence of HBsAg

Of the 1,927 samples included in the study, 99 participants were positive for HBsAg. The overall estimated prevalence was 4.2% (95% CI: 2.7–6.3) after taking the sampling design and weight of each sample into account (Table 2). Males had a higher prevalence of HBsAg (4.8%, 95% CI: 3.5–6.5) than females (3.5%, 95% CI: 1.7–6.9), although there was no statistical significance. Further, the 20–24 (10.7%, 95% CI: 5.0–21.3) and 25–29 (10.4%, 95% CI: 5.5–18.9) year age groups had higher prevalence, and the groups aged less than 15 years showed lower prevalence than other groups, but again the difference was not significant. The Northern region had the highest prevalence (9.0%, 95% CI: 5.0–15.6) of HBsAg, followed by the Central (3.1%, 95% CI: 2.0–4.7) and Southern (1.2%, 95% CI: 0.5–2.8) regions. There were statistical differences in the prevalence among different ethnicities, showing a higher prevalence in Khmu (8.5%, 95% CI: 4.0–17.2) and Hmong (11.2%, 95% CI: 6.6–18.3). Although the prevalence did not statistically differ according to the exposures examined, a higher prevalence was observed in those with an operation history (8.4%, 95% CI: 3.2–20.3) and sharing a razor (11.3%, 95% CI: 5.3–22.6).

## Factors associated with HBsAg positivity

Age groups, place of residence, ethnicity and sharing a razor were significantly associated with HBsAg positivity in bivariable analysis. According to the multivariable analysis, those aged 20–24 years (adjusted odds ratio [AOR]: 2.3, 95% CI: 1.1–4.6), those aged 25–29 years (AOR: 2.7, 95% CI: 1.3–5.6), those from the Northern region (AOR: 2.8, 95% CI: 1.2–6.6), and those who were Khmu (AOR: 3.6, 95% CI: 2.0–6.8) or Hmong (AOR: 5.0, 95% CI: 3.3–7.5) were significantly more likely to be positive for HBsAg (Table 2).

## Seroprevalence of HCV-Ab

Of the 1,927 samples, only 19 tested positive for HCV-Ab. The overall estimated prevalence of HCV-Ab was 1.6% (95% CI: 0.5–5.3) (Table 3). We observed no statistical differences in the prevalence of each variable. Males had a higher HCV-Ab prevalence (2.9%, 95% CI: 0.7–10.7) than did females (0.4%, 95% CI: 0.1–2.4). Only those aged over 30 years had HCV-Ab, and those ≥40 years had the highest prevalence (6.1%, 95% CI: 2.1–16.8). The Southern region had the highest prevalence (3.3%, 95% CI: 0.6–15.3) of HCV-Ab, followed by the Northern (0.7%, 95% CI: 0.2–2.9) and Central (0.5%, 95% CI: 0.2–1.2) regions. Other minor tribes had a higher HCV-Ab prevalence than Laoloum and Hmong (4.7%, 95% CI: 0.6–27.7). Although the prevalence did not show any statistical differences according to the exposures or risk behaviors, those who reported sharing needle for drug injection showed a higher prevalence among exposures (4.8%, 95% CI: 0.8–23.5).

## Discussion

This is the first population-based seroprevalence study to estimate the national prevalence of HBsAg and HCV-Ab in Lao PDR. The overall prevalence of HBsAg was 4.2% and that of HCV-Ab was 1.6%. These estimates are comparable with previous studies in Lao PDR, which studies targeted specific populations [19,21–33] (Table 4).

Although it did not show a significant difference, the prevalence of both HBsAg and HCV-Ab was higher in males than in females, which is similar to other studies [19,22–24,31,33]. This difference may be related to higher sexual risk behaviors in men, such as multiple sexual partners and less use of condoms [39]. However, there has been no systematic collection of data in Lao PDR has shown sex-differences in cultural behaviors related to

**Table 2. Estimated prevalence of HBsAg and its risk factor analysis.**

| | Absolute number | Estimated prevalence | | Bivariate analysis | | | Multivariable analysis | | |
|---|---|---|---|---|---|---|---|---|---|
| | | Weighted % | 95% CI | Crude odds ratio | 95% CI | p-value | Adjusted odds ratio | 95% CI | p-value |
| **Overall** | 1,927 | 4.16 | 2.74–6.26 | | | | | | |
| **Sex** | | | | | | | | | |
| Female | 1,016 | 3.51 | 1.74–6.93 | ref | | | | | |
| Male | 1,001 | 4.81 | 3.53–6.53 | 1.39 | 0.72–2.66 | 0.29 | | | |
| **Age (years)** | | | | | | | | | |
| 1–4 | 342 | 0.15 | 0.03–0.71 | 0.04 | 0.01–0.17 | <0.001 | 0.03 | 0.01–0.14 | <0.001 |
| 5–9 | 179 | 2.84 | 0.76–10.03 | 0.70 | 0.26–1.93 | 0.47 | 0.59 | 0.21–1.62 | 0.30 |
| 10–14 | 120 | 0.79 | 1.43–4.22 | 0.19 | 0.04–1 | 0.05 | 0.21 | 0.04–1.04 | 0.06 |
| 15–19 | 328 | 4.80 | 2.13–10.45 | 1.21 | 0.54–2.71 | 0.62 | 1.22 | 0.62–2.42 | 0.56 |
| 20–24 | 122 | 10.67 | 5.01–21.31 | 2.87 | 1.24–6.63 | 0.02 | 2.28 | 1.13–4.57 | 0.02 |
| 25–29 | 185 | 10.43 | 5.51–18.85 | 2.80 | 1.23–6.37 | 0.02 | 2.67 | 1.27–5.6 | 0.01 |
| 30–34 | 167 | 3.99 | 1.62–9.48 | 1.00 | 0.38–2.66 | 1.00 | 0.97 | 0.45–2.07 | 0.93 |
| 35–39 | 115 | 5.06 | 1.92–12.65 | 1.28 | 0.38–4.33 | 0.67 | 1.50 | 0.59–3.77 | 0.39 |
| 40– | 369 | 3.99 | 2.42–6.53 | ref | | | ref | | |
| **Place of residence (region)** | | | | | | | | | |
| Northern | 525 | 8.97 | 4.98–15.64 | 8.21 | 2.83–23.81 | 0.001 | 2.78 | 1.16–6.64 | 0.02 |
| Central | 956 | 3.07 | 2.00–4.69 | 2.64 | 1.01–6.91 | <0.05 | 2.29 | 0.95–5.53 | 0.06 |
| Southern | 446 | 1.19 | 0.51–2.75 | ref | | | ref | | |
| **Ethnicity** | | | | | | | | | |
| Laoloum | 1,121 | 1.85 | 1.22–2.79 | ref | | | ref | | |
| Khmu | 293 | 8.50 | 4.00–17.16 | 4.93 | 1.91–12.69 | 0.003 | 3.64 | 1.95–6.80 | <0.001 |
| Hmong | 322 | 11.16 | 6.58–18.28 | 6.66 | 3.26–13.59 | <0.001 | 4.95 | 3.28–7.45 | <0.001 |
| Others | 191 | 1.44 | 0.62–3.28 | 0.77 | 0.34–1.75 | 0.51 | 1.65 | 0.30–8.99 | 0.57 |
| **Operation history** | | | | | | | | | |
| Yes | 122 | 8.39 | 3.18–20.33 | 2.23 | 0.99–4.99 | 0.05 | | | |
| No | 1,805 | 3.95 | 2.70–5.73 | ref | | | | | |
| **Blood transfusion history** | | | | | | | | | |
| Yes | 101 | 4.52 | 1.53–12.62 | 1.09 | 0.32–3.64 | 0.89 | | | |
| No | 1,826 | 4.17 | 2.70–6.39 | ref | | | | | |
| **Sharing a toothbrush** | | | | | | | | | |
| Yes | 107 | 6.95 | 3.49–13.33 | 1.77 | 0.88–3.57 | 0.10 | | | |
| No | 1,820 | 4.05 | 2.63–6.20 | ref | | | | | |
| **Tattoo** | | | | | | | | | |
| Yes | 133 | 6.30 | 2.45–15.22 | 1.31 | 0.43–4.04 | 0.61 | | | |
| No | 1,794 | 4.87 | 3.08–7.65 | ref | | | | | |
| **Sharing a razor** | | | | | | | | | |
| Yes | 49 | 11.34 | 5.31–22.60 | 2.59 | 1.03–6.52 | <0.05 | | | |
| No | 1,878 | 4.71 | 2.98–7.37 | ref | | | | | |
| **Sharing needle for drug injection** | | | | | | | | | |
| Yes | 47 | 3.51 | 0.93–12.39 | 0.70 | 0.14–3.48 | 0.64 | | | |
| No | 1,880 | 4.93 | 3.14–7.66 | ref | | | | | |

HBsAg, hepatitis B surface antigen; CI, confidence interval.

**Table 3. Estimated prevalence of HCV-Ab.**

| | Absolute number | Estimated prevalence | |
|---|---|---|---|
| | | Weighted % | 95% CI |
| **Overall** | 1,927 | 1.57 | 0.45–5.30 |
| **Sex** | | | |
| Female | 1,016 | 0.37 | 0.06–2.37 |
| Male | 1,001 | 2.85 | 0.71–10.71 |
| **Age (years)** | | | |
| 1–4 | 342 | 0 | |
| 5–9 | 179 | 0 | |
| 10–14 | 120 | 0 | |
| 15–19 | 328 | 0 | |
| 20–24 | 122 | 0 | |
| 25–29 | 185 | 0 | |
| 30–34 | 167 | 3.78 | 0.45–25.62 |
| 35–39 | 115 | 0.71 | 0.09–5.52 |
| 40– | 369 | 6.10 | 2.05–16.76 |
| **Place of residence (region)** | | | |
| Northern | 525 | 0.72 | 0.18–2.90 |
| Central | 956 | 0.51 | 0.22–1.18 |
| Southern | 446 | 3.27 | 0.63–15.29 |
| **Ethnicity** | | | |
| Laoloum | 1,121 | 0.87 | 0.38–1.96 |
| Khmu | 293 | 0 | |
| Hmong | 322 | 1.14 | 0.20–6.15 |
| Others | 191 | 4.68 | 0.63–27.67 |
| **Operation history** | | | |
| Yes | 122 | 0.90 | 0.10–7.45 |
| No | 1,805 | 1.62 | 0.45–5.62 |
| **Blood transfusion history** | | | |
| Yes | 101 | 0.30 | 0.03–2.84 |
| No | 1,826 | 1.56 | 0.42–5.71 |
| **Sharing a toothbrush** | | | |
| Yes | 107 | 0 | - |
| No | 1,820 | 1.68 | 0.48–5.70 |
| **Tattoo** | | | |
| Yes | 133 | 1.08 | 0.23–4.96 |
| No | 1,794 | 1.99 | 0.56–6.79 |
| **Sharing a razor** | | | |
| Yes | 49 | 0 | - |
| No | 1,878 | 1.97 | 0.57–6.55 |
| **Sharing needle for drug injection** | | | |
| Yes | 47 | 4.78 | 0.81–23.46 |
| No | 1,880 | 1.80 | 0.46–6.75 |

HCV-Ab, hepatitis C antibody; CI, confidence interval.

**Table 4. Seroprevalence studies on HBV and HCV infection in Lao PDR.**

| Study | Population | Location | Number | Prevalence | |
|---|---|---|---|---|---|
| | | | | HBsAg | HCV-Ab |
| Nouanthong et al. (2021) [22] | Voluntary blood donors | Eight provinces | 5,017 | 6.9% | |
| Xaydalasouk et al (2021) [23] | Patients | Saravan province (three district hospitals and one provincial hospital) | 2,463 | 3.8% | |
| Mangkara et al. (2021) [24] | Dentists and dental workers | Vientiane capital | 317 | 5.0% | |
| Xaydalasouk et al (2018) [26] | Female factory workers | Vientiane capital | 400 | 4% ± 1.9% | 1.8% ± 1.3% |
| Evdokimov et al. (2017) [27] | Children (aged 9 to 50 months) | Vientiane, Khammouane, and Boulhikhamxay province | 1,039 | 1.0% | |
| | Mothers | | 1,039 | 7.0% | |
| Choisy et al. (2017) [28] | Pregnant women attending the antenatal clinic | Vientiane capital (Mahosot hospital) | 13,238 | 5.4% (95 CI: 5.1%–5.8%) | |
| Jutavijittum et al. (2016) [29] | Pregnant women attending the antenatal clinic | Vientiane capital (Mother and Child Hospital) | 3,000 | 5.8% | |
| Komada et al. (2015) [30] | Children (5–9 years) | Central region | 911 | 2.1% (95% CI: 0.8–3.4) | |
| | Mothers (15–45 years) | | 911 | 4.1% (95% CI: 2.6–5.5) | |
| Black et al. (2015) [31] | Healthcare workers | Vientiane capital, Huaphan and Boulhikhamxay provinces (three central, two provincial, and eight district hospitals) | 1,128 | 8.0% | 3.9% |
| Xeuatvongsa et al. (2014) [21] | Children (5–9 years) | Nationwide | 965 | 1.7% (95% CI: 0.8%–2.6%) | |
| | Mothers (15–45 years) | | 965 | 2.9% (95% CI: 1.7%–4.2%) | |
| Black et al. (2014) [32] | Infants | Vientiane and Luang Prabang province | 132 | 0.5% | |
| | Preschool children | Huaphan province | 132 | 4.5% | |
| | School children | Luang Prabang, Bolikhamxai, and Savannaket province | 1,689 | 7.9% | |
| | Pregnant women | Luang Prabang and Vientiane province | 388 | 8.2% | |
| Jutavijittum et al. (2014) [33] | Blood donors | Vientiane capital, Vientiane and Bolikhamsay province | 906 | 9.6% | |
| Jutavijittum et al. (2007) [19] | Blood donors | Vientiane capital, Vientiane and Bolikhamsay province | 13,897 | 8.7% | 1.1% |

HBC, hepatitis B; HCV, hepatitis C; PDR, People's Democratic Republic.

transmission, such as tattooing, piercing, or drug injection. Apart from behavioral and cultural gender differences, some research revealed that females usually develop more intense innate, humoral and cellular responses to viral infections, including hepatitis virus, than males [40,41]. In addition, another study reported that female HBV carriers had lower viral loads than male carriers [42], which might have contributed to the higher prevalence of HBsAg in men than in women. Reportedly, women are more likely to clear the virus spontaneously since they have a more efficient innate inflammatory response to HCV infection [43]. Although these previous studies support our findings on the sex-difference in HBsAg and HCV-Ab prevalence, further studies should investigate the association between the infection and local behaviors by gender.

Our findings related to the prevalence of HBsAg according to age differed from those of other studies, which showed that the prevalence increased with age [12,16,23,33]. The vaccination may have protected those less than 15 years of age from the HBV infection since the

vaccine was introduced in 2001 and expanded to all health facilities in 2006 in Lao PDR. However, it is not clear why individuals aged 30 years and more showed a lower HBsAg prevalence than those aged 20–29 years in this study. In general, older people are more likely to be exposed and infected with HBV and to be chronic carriers due to their longer lifetime; thus, age becomes a contributing factor of chronic HBV. A national survey in Lao PDR indicated that those aged 20–29 years were more likely to have multiple sexual partners and less likely to use condoms [39], which could make them highly vulnerable to HBV, explaining the peak in prevalence at 20–29 years. Further investigations are required to identify the association of age and HBV prevalence.

Only two studies previously analyzed HCV-Ab prevalence in different age groups in Lao PDR, and both indicated that those aged 40 years or older had higher prevalence as compared with those aged younger than 40 years [19,31], which is consisstant with our study finding. Several studies in other countries have also indicated that the prevalence of HCV infection increases with age [44–46]. However, there were no positive cases among the participants younger than 30 years in our survey, suggesting that horizontal transmission in adults might be the main route of infection rather than vertical transmission. An active case finding focused among those more than 30 years old and providing highly effective antiviral therapies to those infected with HCV would be able to prevent the transmission and decrease the prevalence in the future [47].

The Northern region showed the highest prevalence of HBsAg (9.0%) over that of the Central (3.1%) and Southern regions (1.2%). A paper analyzing blood donor samples from eight provinces in Lao PDR described a similar regional pattern [22], which could be derived from the differences in sexual practice and vertical transmission although we lack any supporting evidence. Previous studies found that individuals in the Northern region are more likely to be at high risk of contracting human immunodeficiency infection or other sexually transmitted infections due to prostitution and alcohol [48,49]. Furthermore, the differences among those regions could be affected by the differences in ethnicities, with higher HBsAg prevalence, as Khmu and Hmong live mainly in the Northern regions [50]. According to our knowledge, there have been no papers investigating the ethnicity differences of the HBsAg prevalence in Lao PDR. Hmong people have been recognized as having a high prevalence of HBV infection by the studies in the US [51–54] and Thailand [55–57]. Some studies indicated that Khmu people had insufficient knowledge on the prevention of infectious diseases, including HBV infection [45,58]. Although further investigations on the prevalence and risk behaviors in the regions and ethnicity are required, active interventions, including prevention, testing, and treatment, need to be considered targeting the region and those ethnic groups.

The people from the Southern region and those in minor ethnic groups showed a higher prevalence of HCV-Ab than other two regions and other major ethnic groups. According to our knowledge, there have been no papers investigating the regional and ethnicity differences of the HCV prevalence in Lao PDR. Since HCV is transmitted mainly through some risk behaviors, those behaviors could be more active in the people from Southern region and minor ethnic groups. However, no information has been found to suggest the differences in such behaviors in each region in Lao PDR. Further investigations will be required to identify the factors contributing to the regional differences in HCV-Ab prevalence.

The background characteristics of our sampled population were similar to those of another nationwide population-based study, called Lao Social Indicator Survey II (LSIS II), conducted in 2017 [39]. For example, the locations of current residences—north, central, and south—were 27.2%, 49.6% and 23.1% in our survey and 30.8%, 49.9%, and 19.3% in the LSIS II. The education levels attained by women—non, primary school, secondary school or more—were 20.4%, 38.9% and 40.7% in our survey and 16.0%, 43.1% and 40.8% in the LSIS II, respectively.

The LSIS II applied the multistage stratified cluster sampling method and surveyed more than 25,000 people all over the country. Therefore, since our sampled population likely represents the general population in Lao PDR; the estimates of this study are also considered to be highly representative of the population.

There are some limitations in this study. Our estimates among adults might have potential bias because those aged 20–39 years were sampled from parents of 1–2-year-olds. Vertical transmission is the primary route of transmission of viral hepatitis in children [59]. The children whose parents are infected with HBV or HCV might have an increased risk of vertical transmission. However, in our survey, only 1 of 342 pairs of parents and children were infected by HBV, and no vertical transmission was observed in HCV. The rate of vertical transmission ranges from 1%–28% (mean rate:15.7%) with HBV [59–70] and 3%–15% (mean rate: 7.6%) with HCV [59,71–76]. Thus, the influence of that bias was very small in our survey.

DBS instead of whole blood samples were used for laboratory examination for HBsAg and HCV-Ab in this study since the survey included areas with poor access to laboratory facilities. DBS is not the gold standard for measurement; thus, the laboratory results might differ from the ones using whole blood samples. However, several studies have been conducted to evaluate the correlation between those results [77–84], and systematic reviews demonstrated that DBS and whole blood sampling were associated with excellent accuracy and the pooled estimates of the sensitivity and specificity for those markers were higher than 97% [85–87]. The laboratory results in our study are consistent with those of other studies using whole blood samples.

In summary, this nationwide survey estimated the prevalence of HBV and HCV infection among the general population in Lao PDR to be 4.2% and 1.6%, respectively. While the survey was designed to estimate the prevalence in the whole population, it indicated that prevalence might differ by sex, age group, region, and ethnicity. Further studies to identify the causes for these differences need to be conducted, and relevant responses tailored for each target also need to be taken to control the transmission. Our study will hopefully encourage more comprehensive nationwide seroprevalence surveys to monitor the epidemiological situation of multiple diseases, using a simple tool like DBS, since most past studies have focused on only one disease or on specific areas in Lao PDR. Those findings would enable the Ministry of Health, Lao PDR to take more strategic and effective actions for disease prevention and control, including HBV and HCV.

## Supporting information

**S1 Table. Survey result.** This includes age, sex, residential area, HBsAg, and anti-HCV antibody test results.
(XLSX)

**S1 File. Inclusivity in global research.** Additional information regarding the ethical, cultural, and scientific considerations specific to inclusivity in global research.
(DOCX)

**S1 Dataset.**
(XLSX)

## Acknowledgments

We thank Viengphone Khanthamaly, an officer, Influenza Program, U.S. Center for Diseases Control and Prevention, Lao PDR, for her assistance in the survey implementation. We also thank the study participants for understanding the significance of the serosurvey and offering their biospecimens. We also acknowledge the fieldworkers and supervisors from the provincial

and district health offices, Ministry of Health, Lao PDR. We also thank Enago for poof reading the manuscript.

## Author Contributions

**Conceptualization:** Shinsuke Miyano, Chansay Pathammavong, Yasunori Ichimura, Phonethipsavanh Nouanthong, Lauren Franzel, Tae Un Yang, Hendrikus Raaijimakers, Masafumi Funato, Kenichi Komada, Masahiko Hachiya.

**Data curation:** Yasunori Ichimura.

**Formal analysis:** Shinsuke Miyano, Yasunori Ichimura, Masaya Sugiyama.

**Funding acquisition:** Shinsuke Miyano, Chansay Pathammavong, Masaya Sugiyama, Masahiko Hachiya.

**Investigation:** Shinsuke Miyano, Yasunori Ichimura, Masaya Sugiyama, Chankham Tengbriacheu, Bouaphane Khamphaphongphane, Phonethipsavanh Nouanthong, Tomomi Ota, Masahiko Hachiya.

**Methodology:** Shinsuke Miyano, Phonethipsavanh Nouanthong, Lauren Franzel, Tae Un Yang, Hendrikus Raaijimakers, Masafumi Funato, Kenichi Komada, Masahiko Hachiya.

**Project administration:** Shinsuke Miyano, Chansay Pathammavong, Kongxay Phounphenghack, Chankham Tengbriacheu, Bouaphane Khamphaphongphane, Tomomi Ota.

**Resources:** Shinsuke Miyano, Chansay Pathammavong, Kongxay Phounphenghack, Bouaphane Khamphaphongphane, Tomomi Ota.

**Software:** Shinsuke Miyano.

**Supervision:** Shinsuke Miyano, Chansay Pathammavong, Kongxay Phounphenghack, Chankham Tengbriacheu, Bouaphane Khamphaphongphane, Phonethipsavanh Nouanthong, Tomomi Ota, Masahiko Hachiya.

**Validation:** Shinsuke Miyano, Yasunori Ichimura, Kongxay Phounphenghack, Chankham Tengbriacheu, Bouaphane Khamphaphongphane, Phonethipsavanh Nouanthong, Tomomi Ota, Masahiko Hachiya.

**Visualization:** Shinsuke Miyano.

**Writing – original draft:** Shinsuke Miyano.

**Writing – review & editing:** Chansay Pathammavong, Yasunori Ichimura, Kongxay Phounphenghack, Chankham Tengbriacheu, Bouaphane Khamphaphongphane, Phonethipsavanh Nouanthong, Lauren Franzel, Tae Un Yang, Hendrikus Raaijimakers, Tomomi Ota, Masafumi Funato, Kenichi Komada, Masahiko Hachiya.

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
