## [Decision Letter · Decision Letter 0]

24 Aug 2022

PONE-D-22-13251Prevalence of hepatitis B and C virus infections in Lao People’s Democratic Republic: a population-based cross-sectional surveyPLOS ONE

Dear Dr. Miyano,

Thank you for submitting your manuscript to PLOS ONE. After careful consideration, we feel that it has merit but does not fully meet PLOS ONE’s publication criteria as it currently stands. Therefore, we invite you to submit a revised version of the manuscript that addresses the points raised during the review process.

We look forward to receiving your revised manuscript.

Kind regards,

Livia Melo Villar

Academic Editor

PLOS ONE

Journal Requirements:

“This work was supported by the Grants for the National Center for Global Health and Medicine [grant number 19A01] and the Grant for the National Immunization Program [FY2019], Lao PDR. However, the funding source was not involved in design of the study, collection, analysis, and interpretation of data, or writing of the manuscript.”

Additional Editor Comments:

Dear Author,

Thanks for sending the paper intitled: "Prevalence of hepatitis B and C virus infections in Lao People’s Democratic Republic: a population-based cross-sectional survey". This is the first population-based prevalence investigation of hepatitis B and C viruses in the Lao People's Democratic Republic. The study is significant however some issues were raised by reviewers and I suggest major revision of this article.

Sincerely,

Livia

Reviewers' comments:

Reviewer's Responses to Questions

**Comments to the Author**

1. Is the manuscript technically sound, and do the data support the conclusions?

Reviewer #1: Yes

Reviewer #2: Yes

2. Has the statistical analysis been performed appropriately and rigorously? 

Reviewer #1: Yes

Reviewer #2: Yes

3. Have the authors made all data underlying the findings in their manuscript fully available?

Reviewer #1: Yes

Reviewer #2: Yes

4. Is the manuscript presented in an intelligible fashion and written in standard English?

Reviewer #1: Yes

Reviewer #2: Yes

5. Review Comments to the Author

Reviewer #1: Dear authors,

The paper entitled has has merit and will probably give a significant contribution to the field. However, some improvements should be made to make it suitable for publication.

-Table 1 is disorganized and needs to be better structured. For example, it would be better to have separate columns for the studied population and the "n" sample. Furthermore, it would be better to use it in comparison with the results obtained in the study rather than in the introduction.

Methods

-Lines 129-131: In the sentence: “For the third stage, 42 participants 130 (including 8, 6, 6, 16, and 6 in the 1–2, 5–14, 15–19, 20–39, and ≥40 year age groups, 131 respectively) were randomly selected...” What do these numbers mean? Please, clarify.

Results:

-In general, tables could be better structured and organized. Table 2, for exemple, could be summarized by classifying age groups every 10 years.

-Page 18, line 219: “those WHO REPORTED sharing needle for drug injection showed...”

-As no variables were statistically significant, table 4 is not necessary.

Discussion

Page 20, line 238: “In addition, female HBV carriers have lower viral loads than male carriers...” The authors did not performed viral load analysis in this study. If this comment refers to another study (reference 42), this needs to be clarified.

Page 21, lines 245-255: Could this difference be related to the sexual behavior of individuals aged 20-29 years? Are they more likely to be single? Multiple partners? Non condom users?

Page 22, line 267: “...the differences in sexual practice and parent-to-child transmission...” it would be better to replace "parent-to-child" with "vertical transmission".

Page 22, lines 275-277: The sentences “Hmong people have been recognized as having a high prevalence of HBV infection by the studies in the US [49-52] and Thailand [53-55] due to their sexual risk behaviors.” And “Some papers indicated that Khmu people had higher likeliness of aspiration to the commercial sex industry due to their economic restriction...” Please check if these affirmations complies with ethical criteria, as associating an ethnicity with risky sexual behavior can give rise to discrimination. I suggest rewriting these sentences observing this criteria.

Reviewer #2: The study presented by Miyano et al is the first population-based prevalence investigation of hepatitis B and C viruses in the Lao People's Democratic Republic. The study is significant since it is the first population-based prevalence study in this country, bringing new information to the region. Despite its merits, the article contains some major aspects that must be corrected and clarified:

- Title:

To highlight the significance of the work, the authors should mention in the title that it is the first population-based prevalence study made in the country.

- Abstract:

Line 44 - phrase is not clear, needs adjustment

- Instroduction

Line 89 - phrase is not clear, needs adjustment

- Materials and methods:

Section "Sample size calculation"

Lines 112 to 120 - We know that the authors used a sample size from the rubeola and measles studies, but it would be clearer and more straightforward to begin with the sample size for hepatitis B and C.

Line 130 - Why were the age groups divided in this way? Is there a reason? Was it based on the population's average age? Please elaborate on this point.

- Results:

Line 169 - Is the population based on the rubella and measles survey? or is it the population already estimated for hepatitis B and C? This is not very clear.

Line 171 - The same goes for this sentence, it is noticed here that the population examined is larger than the estimated sample n for HBV and HCV. Is this population based on the rubella and measles survey, or on the estimated population for hepatitis? Please clarify.

Table 2 - The sum of the individuals in this "Sex" category does not match the total of the analyzed population, nine individuals are missing. Are they missing their sex? Provide the missing data in the table, either as a new column or footnote identifiable by a marker.

Table 2 - The same occurs for the ethnicity category, 73 individuals are missing in the sum to reach the total value. And also in the "Education history" category, 199 individuals are missing for the total sum of the population.

In the paragraph of the section "Seroprevalence of HBsAg" (lines 182 to 195) and "Seroprevalence of HCV-Ab" (lines 210 o 220), provide the absolute number of the number of cases for the data presented.

Table 3 and 4 - Provide the absolute number of cases for the data presented.

- Discussion:

Lines 239 to 240 - Because the study offered as a comparison was conducted on a relatively specific group, the placement should be interpreted with caution. Studies in different communities and areas may demonstrate different or equal prevalences for HCV, where women may have a greater or equal soprevalence for HCV than males. (Niu et al, 2016 [DOI: 10.1186/s40064-016-3224-z]; Peliganga et al, 2021 [DOI: 10.3390/pathogens10121633]; Rangel et al, 2021[DOI: 10.1016/j.clgc.2020.08.006]; Bisseye et al, 2018).

Lines 258 to 259 - The placement presented was unclear. However, there were no cases in adults under the age of 30, and was maternal-infant transmission suspected? Did you mean to mean those over the age of 40? In addition, the fact that the prevalence of hepatitis C grows with age, as studies have demonstrated, is an interesting approach to consider and discuss. (Peliganga et al, 2021 [DOI: 10.3390/pathogens10121633]; Abdella, et al, 2020 [DOI: 10.1371/journal.pone.0241086]; Vermeulen et al, 2017 [DOI: 10.3201/eid2309.161594]).

Lines 269 to 270 - The term used is out of date and should be replaced by sexually transmitted infections.

6. PLOS authors have the option to publish the peer review history of their article (what does this mean?). If published, this will include your full peer review and any attached files.

Reviewer #1: No

Reviewer #2: No

---

## [Author Response · Author response to Decision Letter 0]

14 Sep 2022

Reviewer #1: 

1. Table 1 is disorganized and needs to be better structured. For example, it would be better to have separate columns for the studied population and the "n" sample. Furthermore, it would be better to use it in comparison with the results obtained in the study rather than in the introduction.

We appreciate the reviewer's suggestion. We have re-organized Table 1 by adding the columns for “study location” and “number of the population”. In addition, we have relocated this Table to the Discussion section and put it as Table 4.

2. Lines 129-131: In the sentence: “For the third stage, 42 participants (including 8, 6, 6, 16, and 6 in the 1–2, 5–14, 15–19, 20–39, and ≥40 year age groups, respectively) were randomly selected...” What do these numbers mean? Please, clarify.

We appreciate the reviewer's clarification. Due to the stratification of age groups, we randomly selected 42 participants per village which included 8 participants in 1–2 year, 6 participants in 5–14 year, 6 participants in 15–19 year, 16 participants in 20–39 year, and 6 participants in ≥40 year age groups, respectively. We revised this sentence as we wrote here.

3. Results: In general, tables could be better structured and organized. Table 2, for exemple, could be summarized by classifying age groups every 10 years.

We appreciate the reviewer's suggestion in Table 2. We have revised the table, especially age groups, as the reviewer suggested.

4. Page 18, line 219: “those WHO REPORTED sharing needle for drug injection showed...”

We appreciate the reviewer's pointing out our writing error. We have revised the sentence as the reviewer suggested.

5. As no variables were statistically significant, table 4 is not necessary.

We appreciate the reviewer's suggestion in Table 4. Although we understand the reviewer's reasonable suggestion, for the transparency of the results/data, we would like the reviewer to accept that we keep Table 4.

6. Page 20, line 238: “In addition, female HBV carriers have lower viral loads than male carriers...” The authors did not perform viral load analysis in this study. If this comment refers to another study (reference 42), this needs to be clarified.

We appreciate the reviewer's clarification. We have rephrased the sentence like, “In addition, another study reported that female HBV carriers had lower viral loads than male carriers [42], which might have contributed to the higher prevalence of HBsAg in men than in women.”

7. Page 21, lines 245-255: Could this difference be related to the sexual behavior of individuals aged 20-29 years? Are they more likely to be single? Multiple partners? Non condom users?

We appreciate the reviewer's important comments. We found the reference (the Lao Social Indicators Survey II (LSIS II) in 2017) showing that those aged 20-29 years were more likely to have multiple partners and less likely to use condoms. We added an assumption on relation between sexual behaviour and the higher prevalence among those aged 20-29 years as the reviewer suggested.

8. Page 22, line 267: “...the differences in sexual practice and parent-to-child transmission...” it would be better to replace "parent-to-child" with "vertical transmission".

We appreciate the reviewer's suggestion. We have replaced "parent-to-child" with "vertical transmission" as the reviewer suggested.

9. Page 22, lines 275-277: The sentences “Hmong people have been recognized as having a high prevalence of HBV infection by the studies in the US [49-52] and Thailand [53-55] due to their sexual risk behaviors.” And “Some papers indicated that Khmu people had higher likeliness of aspiration to the commercial sex industry due to their economic restriction...” Please check if these affirmations comply with ethical criteria, as associating an ethnicity with risky sexual behavior can give rise to discrimination. I suggest rewriting these sentences observing these criteria.

We appreciate the reviewer's very important suggestion. We agree that we need to be more sensitive on such a matter. We decided to write just the epidemiological fact from other papers and remove the parts mentioning the association between ethnicity and risky sexual behavior to avoid rising discrimination.

Reviewer #2: 

1. Title: To highlight the significance of the work, the authors should mention in the title that it is the first population-based prevalence study made in the country.

We appreciate the reviewer's very positive suggestion. We have revised the title as the reviewer recommended.

2. Abstract:　Line 44 - phrase is not clear, needs adjustment

We appreciate the reviewer's suggestion. We have rewritten the phrase to make it clear by adding the cut-off value for both HBsAg and HCV-Ab.

3. Introduction：Line 89 - phrase is not clear, needs adjustment

We appreciate the reviewer's suggestion. This phrase wanted to indicate that there is no seroprevalence studies in the general population since the first prevalence survey in 2012 covered only children and their mothers. We have rewritten the phrase to make it clear. 

4. Materials and methods:　Section "Sample size calculation"　Lines 112 to 120 - We know that the authors used a sample size from the rubeola and measles studies, but it would be clearer and more straightforward to begin with the sample size for hepatitis B and C.

We appreciate the reviewer's suggestion. We began this part straightforward with the sample size for hepatitis B and C and removed the sample size calculation for measles and rubella. We have revised this part as the reviewer recommended.

5. Materials and methods: Line 130 - Why were the age groups divided in this way? Is there a reason? Was it based on the population's average age? Please elaborate on this point.

We appreciate the reviewer's clarification. Since the primary objective of this survey was to compare the seroprevalence of measles and rubella between 2014 and 2019, this survey design required to be the same with the one in 2014. In 2014, the age groups were divided based on the exposure history to routine immunization services and supplementary immunization activities (SIAs) services. We stratified the age groups for strategic sampling and estimated the prevalence in each age group. That is the reason why the age groups were divided in this way.

6. Results: Line 169 - Is the population based on the rubella and measles survey? or is it the population already estimated for hepatitis B and C? This is not very clear.

We appreciate the reviewer's clarification. The numbers are absolute numbers of blood samples. We have collected 2,043 samples in the survey. After excluding inappropriate samples, 1,927 samples were tested for both measles/rubella and hepatitis B/C. While the number of the samples was less than required sample size for measles/rubella prevalence estimation (88.2% of the required sample size, which was 2,184), it was more than required sample size for hep B/C prevalence estimation (103.0% of the required sample size, which was 1,872). To avoid the confusion, we have removed the “93.5% of required samples size” since this information is based on the required samples size for the measles/rubella survey and added the percentage (103.0%) against the required sample size for hepatis B/C (1,872 samples).

7. Results: Line 171 - The same goes for this sentence, it is noticed here that the population examined is larger than the estimated sample n for HBV and HCV. Is this population based on the rubella and measles survey, or on the estimated population for hepatitis? Please clarify.

We appreciate the reviewer's clarification. As mentioned above, the numbers are absolute numbers of blood samples. We have collected 2,043 samples in the survey. After excluding inappropriate samples, 1,927 samples were tested for both measles/rubella and hepatitis B/C. While the number of the samples was less than required sample size for measles/rubella prevalence estimation (88.2% of the required sample size, which was 2,184), it was more than required sample size for hep B/C prevalence estimation (103.0% of the required sample size, which was 1,872). To avoid the confusion, we have removed the “93.5% of required samples size” since this information is based on the required samples size for the measles/rubella survey and added the percentage (103.0%) against the required sample size for hepatis B/C (1,872 samples).

8. Table 2 - The sum of the individuals in this "Sex" category does not match the total of the analyzed population, nine individuals are missing. Are they missing their sex? Provide the missing data in the table, either as a new column or footnote identifiable by a marker.

We appreciate the reviewer's pointing out our error. The number was our typo. We have corrected the number in Table 2.

9. Table 2 - The same occurs for the ethnicity category, 3 individuals are missing in the sum to reach the total value. And also in the "Education history" category, 199 individuals are missing for the total sum of the population.

We appreciate the reviewer's pointing out our error. The number for Ethnicity category was our typo. We have corrected the number in Table 2 (no missing for the ethnicity category). 

The information on the Education History of 199 individuals were missing. We have added the footnote for the information.

10. In the paragraph of the section "Seroprevalence of HBsAg" (lines 182 to 195) and "Seroprevalence of HCV-Ab" (lines 210 o 220), provide the absolute number of the number of cases for the data presented.

We appreciate the reviewer's comment on this point. As we mentioned in the Methodology section, the survey results should be demonstrated in "estimated prevalence (95% CI)", considering the sampling weight of each sample to elicit representative and unbiased results. Showing "absolute numbers" in the Result section might confuse the readers. Instead, we have added the absolute numbers in Table 3 and 4. We would appreciate if the reviewer could accept our response.

11. Table 3 and 4 - Provide the absolute number of cases for the data presented.

We appreciate the reviewer's suggestion in Table 3 and 4. We have added the absolute numbers of cases for the date presented in Table 3 and 4 as the reviewer suggested.

12. Discussion: Lines 239 to 240 - Because the study offered as a comparison was conducted on a relatively specific group, the placement should be interpreted with caution. Studies in different communities and areas may demonstrate different or equal prevalences for HCV, where women may have a greater or equal soprevalence for HCV than males. (Niu et al, 2016 [DOI: 10.1186/s40064-016-3224-z]; Peliganga et al, 2021 [DOI: 10.3390/pathogens10121633]; Rangel et al, 2021[DOI: 10.1016/j.clgc.2020.08.006]; Bisseye et al, 2018).

We appreciate the reviewer's very important comment, together with some helpful reference papers. We have agreed on this point and decided to remove the sentence related to the epidemiology in gender to avoid misinterpretation. 

13. Discussion: Lines 258 to 259 - The placement presented was unclear. However, there were no cases in adults under the age of 30, and was maternal-infant transmission suspected? Did you mean to mean those over the age of 40? In addition, the fact that the prevalence of hepatitis C grows with age, as studies have demonstrated, is an interesting approach to consider and discuss. (Peliganga et al, 2021 [DOI: 10.3390/pathogens10121633]; Abdella, et al, 2020 [DOI: 10.1371/journal.pone.0241086]; Vermeulen et al, 2017 [DOI: 10.3201/eid2309.161594]).

We appreciate the reviewer's clarification. Since our survey did not find any positive cases in younger age group (under 30 years old), we assumed that the transmission might be circulated mainly among adults (horizontal transmission) rather than mother-to-child (vertical transmission).

We also appreciate the reviewer’s suggestion on the discussion point with some reference papers. We have added the sentence, using those papers as references.

14. Discussion: Lines 269 to 270 - The term used is out of date and should be replaced by sexually transmitted infections.

We appreciate the reviewer's suggestion. We have replaced the term as the reviewer recommended.

---

## [Decision Letter · Decision Letter 1]

27 Oct 2022

PONE-D-22-13251R1Prevalence of hepatitis B and C virus infections in Lao People’s Democratic Republic: the first national population-based cross-sectional surveyPLOS ONE

Dear Dr. Miyano,

Thank you for submitting your manuscript to PLOS ONE. After careful consideration, we feel that it has merit but does not fully meet PLOS ONE’s publication criteria as it currently stands. Therefore, we invite you to submit a revised version of the manuscript that addresses the points raised during the review process.

We look forward to receiving your revised manuscript.

Kind regards,

Livia Melo Villar

Academic Editor

PLOS ONE

Journal Requirements:

Additional Editor Comments :

Dear Author,

Thanks for sending the revision of this manuscript. After reading the comments of reviewers, I suggested minor revision,

Sincerely,

Livia Villar

Reviewers' comments:

Reviewer's Responses to Questions

**Comments to the Author**

1. If the authors have adequately addressed your comments raised in a previous round of review and you feel that this manuscript is now acceptable for publication, you may indicate that here to bypass the “Comments to the Author” section, enter your conflict of interest statement in the “Confidential to Editor” section, and submit your "Accept" recommendation.

Reviewer #1: All comments have been addressed

Reviewer #2: All comments have been addressed

2. Is the manuscript technically sound, and do the data support the conclusions?

Reviewer #1: Yes

Reviewer #2: Yes

3. Has the statistical analysis been performed appropriately and rigorously? 

Reviewer #1: I Don't Know

Reviewer #2: Yes

4. Have the authors made all data underlying the findings in their manuscript fully available?

Reviewer #1: Yes

Reviewer #2: Yes

5. Is the manuscript presented in an intelligible fashion and written in standard English?

Reviewer #1: Yes

Reviewer #2: Yes

6. Review Comments to the Author

Reviewer #1: Dear authors,

The manuscript was improved considerably. Most of my queries were addressed. However, I would suggest that a careful review of the language be carried out before publication.

Reviewer #2: I don't have any other observations. All comments have been addressed.

I congratulate the authors for their work.

7. PLOS authors have the option to publish the peer review history of their article (what does this mean?). If published, this will include your full peer review and any attached files.

Reviewer #1: No

Reviewer #2: No

---

## [Author Response · Author response to Decision Letter 1]

28 Oct 2022

Reviewer #1: 

Dear authors,

The manuscript was improved considerably. Most of my queries were addressed. However, I would suggest that a careful review of the language be carried out before publication.

We appreciate the reviewer's suggestion. We requested English proofreading by native English speakers and made some corrections and revisions.

---

## [Editor Report · Decision Letter 2]

24 Nov 2022

Prevalence of hepatitis B and C virus infections in Lao People’s Democratic Republic: the first national population-based cross-sectional survey

PONE-D-22-13251R2

Dear Dr. Miyano,

We’re pleased to inform you that your manuscript has been judged scientifically suitable for publication and will be formally accepted for publication once it meets all outstanding technical requirements.

Kind regards,

Livia Melo Villar

Academic Editor

PLOS ONE

Additional Editor Comments (optional):

Dear Author,

Thanks for sending me the paper for my evaluation. After reading reviewer's comments and author's response, I recommend the publication of this paper,

sincerely,

Livia
---

## [Editor Report · Acceptance letter]

20 Dec 2022

PONE-D-22-13251R2 

Prevalence of hepatitis B and C virus infections in Lao People’s Democratic Republic: the first national population-based cross-sectional survey 

Dear Dr. Miyano:

I'm pleased to inform you that your manuscript has been deemed suitable for publication in PLOS ONE. Congratulations! Your manuscript is now with our production department. 

Kind regards, 

on behalf of

Dr. Livia Melo Villar 

Academic Editor

PLOS ONE